# Parametric Analysis of Nonlinear Oscillations of the "Rotor–Weakly Conductive Viscous Fluid–Foundation" System under the Action of a Magnetic Field

**Almatbek Kydyrbekuly [1], Algazy Zhauyt [2] and Gulama-Garip Alisher Ibrayev [1],***

[1] Department of Mechanics, Al-Farabi Kazakh National University, Almaty 050040, Kazakhstan; almatbek@list.ru
[2] Department of Electronics and Robotics, Almaty University of Power Engineering and Telecommunications Named after G. Daukeyev, Almaty 050013, Kazakhstan; a.zhauyt@aues.kz
* Correspondence: ybraev.alysher@mail.ru; Tel.: +7-327-7051111745

**Abstract:** The generalized dynamic model of the rotor system, presented in this paper, is the first model that takes into account the interconnected oscillations of the "rotor–weakly conductive fluid–foundation" system under the action of parameters such as fluid and rotor motion, linear eccentricity, friction forces, foundation vibration and nonlinear characteristics of rolling bearings, as well as the action of a magnetic field on the fluid. Consistent equations of motion for the "rotor–weakly conductive fluid–foundation" system were derived and solved analytically. Forced and natural oscillations of the system were analyzed, and the distinctive features of the rotor system dynamics were revealed. The values of frequencies and amplitudes, which are one of the main factors determining the dynamic behavior of the system, were obtained and studied.

**Keywords:** rotor; moving foundation; nonlinear oscillations; critical frequency; natural oscillations; weakly conductive liquid; magnetic and electrohydrodynamics

## 1. Introduction

In many theoretical and practical studies on the dynamics of rotor systems containing liquid, only oscillations of the rotor with liquid are considered, without taking into account the electromagnetic properties of the liquid and the mobility of the foundation [1]. This assumption leads to certain errors in assessing the dynamic and kinematic characteristics of the rotor system [2]. Studies of dynamic systems such as rotary systems show the importance of taking into account the electromagnetic properties of the fluid, the nonlinear properties of the shaft supports, foundation vibration and the need to develop measures to reduce them [3,4]. The intensive development of magnetic and electrohydrodynamics (hereinafter referred to as MEHD) started in the 1960s by the Melcher group in the USA. In Europe, these studies were started by French and Spanish scientific centers [5] and others [6], where the issues of the possibility of using electrohydrodynamic effects in production and the importance of developing a general theory of MEHD were considered. In 1966, Taylor was the first to show how the application of a uniform electric field can deform the shape of a weakly conductive liquid depending on its electrohydrodynamic properties [7]. In the USSR, this direction was developed by authors such as V.V. Gogosov and I.E. Tarapov [8]. As a result, already in the 1970–1980s, the main provisions and systems of MEHD equations were developed, and the conditions under which various MEHD models are valid were considered. Intensive MEHD studies using applied physics methods were carried out in [9], where gas-dynamic flows with similarly charged particles were studied and new directions in MEHD were analyzed: MEHD turbulence, unsteady effects, and new methods for diagnosing MEHD flows. Some MEHD problems on the stability of an inhomogeneously heated low-conductivity liquid were studied in [10]. A

large amount of work on the study of electrophysical processes in dielectric liquids initiated by high-intensity fields was carried out by the authors of [11]. The results of recent MEHD studies are presented in [12].

Oscillations of the free surface of a viscous fluid in a rotating cylindrical vessel and stability of rotational motions were first considered by Stewartson [13] and Kostandyan; further, their studies were continued in the works of Bauer [14], Kimura [15], and Eidel. In these papers, linearized equations of a system acting on a vessel from the liquid side are studied, where it is assumed that the free surface of the liquid differs little from the unperturbed cylindrical shape. A similar formulation of the problem and a similar mathematical model was used in [14–17], where the oscillations of a rotating cylinder partially filled with an ideal and viscous fluid were studied.

This problem originates in the works of Kelvin (1877, 1880), Lamb (1945), Zhukovsky (1948, 1949) and Chetaev (1957), and was first solved in a general form by B.I. Rabinovich in 1951 [18]. In 1952, N.N. Moiseev, independent of [18], obtained similar equations of motion, which can be found in more detail in [19]. Among the works of modern authors devoted to various aspects of this direction, we should mention the following papers: in [20], the analytical and numerical results obtained in the study of the motion of a system consisting of a rigid body with a cavity filled with a viscous fluid are presented; in [21–23], chaotic motions of a rigid body and with a cavity filled with a liquid are studied; in [24], the experimental work was carried out to study the oscillations and displacements of the free surface of a liquid for the case when the cavity is filled with two different liquids; in [25], a quasi-analytical model of oscillations of magnetic fluids under low gravity under the action of external inhomogeneous magnetic fields is presented [26], and the problem of free and forced oscillations is solved for models with axisymmetric geometries and loads in a linear formulation by the Ritz and finite element methods.

The analysis of the studies of rotor systems with a cavity partially filled with liquid shows that the electromagnetic properties of the liquid were taken into account only in a few works [27]. Therefore, the study of the problem where the effect of a magnetic field on the oscillation of an oscillatory system is evaluated becomes especially important. The solution to this problem is complicated by the fact that the motion of a rotating rotor and the motion of a weakly conductive viscous fluid in its cavity are interconnected under the action of the electromagnetic field. The system being solved includes the equations of movement of a solid body, the equations of a continuous medium and boundary conditions for the liquid [28,29].

## 2. Materials and Methods

The rotor system rotates on rolling bearings (see Figure 1). In this case, elastic deformations in the rolling bearing occur in the radial and axial directions and are nonlinear [1]. The radial compliance of the bearings is caused by the deformation of the rolling elements and roller ways at the points of contact [2]. Consider a symmetrical vertical rotor of mass $m$, having a cylindrical cavity of radius $R$, and a static imbalance $e$. A cylinder of height $h$ is partially filled with a weakly conductive viscous liquid. The angular velocity of the rotor (shaft) $\Omega_0 = const\,\pi$ is considered sufficiently large as it is beyond its critical velocity.

The equations of the bearing static equilibrium are compiled in accordance with the Hertz theory [26–28]. The rolling bearing has a nonlinear stiffness characteristic of the type:

$$F_C = c_0\delta_r + c_1\delta_r^3$$

where $F_C$ is a component of the restoring force in the radial direction, $\delta_r$ is the deformation in the radial direction, $c_0$ and $c_1$ are stiffness coefficients for the linear and cubic terms.

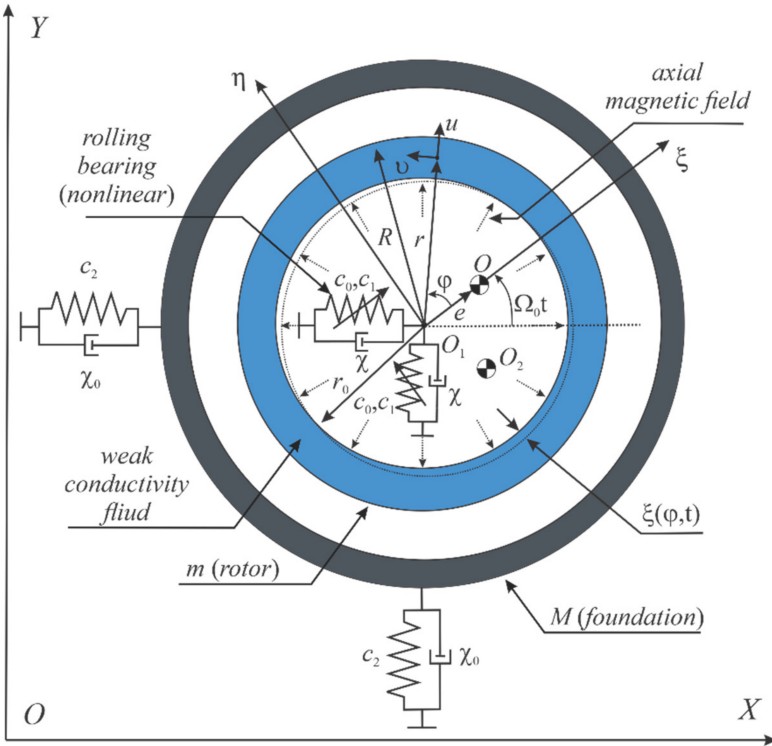

**Figure 1.** Scheme of a rotor partially filled with a weakly conductive liquid mounted on an elastic foundation.

The outer ring is rigidly connected to the foundation of mass $M$, which is mounted on an elastic support, with a linear stiffness coefficient $c_2$. To compile the equation of motion of the system, a fixed coordinate system $OXY$ is introduced. In an equilibrium state, the geometric center of the shaft (rotor) and the center of gravity of the foundation coincide with the origin of the fixed coordinate system [3]. The coordinates in the displaced position of the center of the shaft (rotor) $O_1$ are denoted by $x_1$ and $y_1$, and the center of gravity of the rotor is denoted by $x$ and $y$. The coordinates of the center of gravity of the foundation $O_2$ are denoted by $x_2$ and $y_2$, $c_0$ and $c_1$ are the coefficients of rigidity of the rotor support (rigidity of the rolling bearing), $\chi$ and $\chi_0$ are the coefficients of external friction [4–6]. It is assumed that the rotor performs a plane-parallel motion, and there is no rotation of the foundation around the coordinate axes. In this case, the equations of motion of the system are written as:

$$\left.\begin{array}{l} m\ddot{x} + 2c_0(x - x_2) + 2c_1(x - x_2)^3 + \chi\dot{x} = me\Omega_0^2\cos\Omega_0 t + F_x \\ m\ddot{y} + 2c_0(y - y_2) + 2c_1(y - y_2)^3 + \chi\dot{y} = me\Omega_0^2\sin\Omega_0 t + F_y \\ m\ddot{x}_2 + 2c_2 x_2 - 2c_0(x - x_2) - 2c_1(x - x_2)^3 + \chi_0\dot{x}_2 = 0 \\ m\ddot{y}_2 + 2c_2 y_2 - 2c_0(x - y_2) - 2c_1(y - y_2)^3 + \chi_0\dot{y}_2 = 0 \end{array}\right\} \tag{1}$$

where $F_x$ and $F_y$ are components of the fluid reaction force:

$$F_x = Rh\int_0^{2\pi}\sigma_n\Big|_{r=R}\cos(\Omega_0 t + \phi)d\phi \tag{2}$$

$$F_y = Rh\int_0^{2\pi}\sigma_n\Big|_{r=R}\sin(\Omega_0 t + \phi)d\phi \tag{3}$$

where $h$ is the height of the rotor cavity and $\sigma_n\,|_{r=R}$ is the normal pressure of the viscous fluid on the rotor wall.

The interaction of the azimuthal magnetic field induced in the liquid with radial and axial currents creates magnetic forces that deform the free surface of the liquid along the axis [7]. For a liquid with low conductivity, the magnetic Reynolds number is $Re_m << 1$ at $\sigma << 1$ and is unchanged in other scales. In this case, the induced magnetic field can be neglected in comparison with the applied external magnetic field, and the magnetic induction equation is not considered [9]. In the first approximation, the induced currents can be considered radial, and the influence of axial currents can be neglected.

Then, the Lorentz force vector is written as:

$$\vec{F}_L = \sigma_F \mu_F^2 H^2 \overline{V} \tag{4}$$

where $\sigma_F$, $\mu_F$, and $H$ are electrical conductivity, magnetic permeability and intensity of a constant magnetic field; $\vec{V}$ is the velocity vector of a fluid particle in a moving coordinate system.

Taking into account (4), the differential equations of motion of a weakly conductive viscous fluid in a polar coordinate system rotating together with the rotor are written as:

$$
\begin{aligned}
\frac{\partial u}{\partial t} - 2\Omega_0 v + u\frac{\partial u}{\partial r} + \frac{v}{r}\frac{\partial u}{\partial \phi} - \frac{v^2}{r} - v\frac{\partial f}{r\partial \phi} + b_m u &= -\frac{1}{\rho}\frac{\partial P}{\partial r} - \ddot{x}\cos(\Omega_0 t + \phi) - \ddot{y}\sin(\Omega_0 t + \phi) \\
\frac{\partial v}{\partial t} + 2\Omega_0 u + u\frac{\partial v}{\partial r} + \frac{v}{r}\frac{\partial v}{\partial \phi} + \frac{uv}{r} + v\frac{\partial f}{\partial r} + b_m v &= -\frac{1}{\rho r}\frac{\partial P}{\partial \phi} + \ddot{x}\sin(\Omega_0 t + \phi) - \ddot{y}\cos(\Omega_0 t + \phi)
\end{aligned}
\tag{5}
$$

where $P$, $u$, and $v$ are the pressure, radial and tangential velocity components of a liquid particle in a polar coordinate system, $b_m = \frac{\sigma_F \mu_f^2 H^2}{\rho}$ is a parameter that determines the effect of a magnetic field on a liquid particle; $\rho$ and $\nu$ are the density and the coefficient of the kinematic viscosity of the liquid; $f$ is the Laplacian of the stream function [10–12], i.e.,

$$f = \frac{1}{r}\frac{\partial u}{\partial \phi} - \frac{\partial v}{\partial r} - \frac{v}{r} \tag{6}$$

The continuity equation for $\rho$ = const:

$$\frac{\partial(ur)}{\partial r} + \frac{\partial v}{\partial \phi} = 0 \tag{7}$$

The boundary conditions of the hydrodynamic problem have the following form on the wall of the rotor:

$$u|_{r=R} = 0$$

$$u|_{r=R} = 0 \tag{8}$$

On the free surface of the liquid:

$$v\rho\left(\frac{1}{r}\frac{\partial u}{\partial \phi} + \frac{\partial v}{\partial r} - \frac{v}{r}\right)\bigg|_{r=r_0} = 0 \tag{9}$$

$$\left[-P + \frac{1}{2}\rho\Omega_0^2\left(r^2 - r_0^2\right) + 2v\rho\frac{\partial u}{\partial r}\right]\bigg|_{r=r_0+\xi(\phi,t)} = 0 \tag{10}$$

where $\xi(\varphi, t)$ is the displacement of the free surface of a liquid from an equilibrium position:

$$\frac{\partial \xi(\phi, t)}{\partial t} = u|_{r=r_0} = 0 \tag{11}$$

Excluding the unknown pressure $P$ from system Equation (5), taking into account Equations (6) and (7), we obtain:

$$\frac{\partial f}{\partial t} + bf - \nu\Delta f = -\left(u\frac{\partial f}{\partial r} + \frac{v}{r}\frac{\partial f}{\partial \phi}\right) \tag{12}$$

where $\Delta$ is the Laplace operator in the polar coordinate system.

Equation (12) is solved by the method of successive approximations [30]. In the equilibrium position of the rotor and the foundation, i.e., in the absence of oscillations of the system [31], the viscous fluid rotates with the rotor as a single solid body. Then:

$$u_0 = 0, \ v_0 = 0, \ f_0 = 0 \tag{13}$$

Taking into account Equation (12), Equation (13) in the first approximation takes the form:

$$\frac{\partial f_1}{\partial t} + bf_1 - \nu\Delta f_1 = 0 \tag{14}$$

Further, for convenience of notation, we omit the index 1 in Equation (14), as well as in the functions $u_1$, $v_1$, and $P_1$ in the first approximation [32,33].

The continuity equation and the boundary conditions of the hydrodynamic problem in the first approximation are determined by Equations (7)–(11). Differential Equations (1) and (14) with boundary conditions Equations (7)–(11) are consistent equations of motion of the rotor, foundation and weakly conductive viscous fluid [34]. To calculate the hydrodynamic force, the movement of the rotor and the foundation, it is convenient to represent them on the complex plane in the following form:

$$z = Ae^{i\Omega_0 t} + Be^{i\omega t} \tag{15}$$

$$z_2 = Ce^{i\Omega_0 t} + De^{i\omega t} \tag{16}$$

Taking Equations (14) and (15) into account, the equations of motion of a viscous fluid Equation (5) take the form:

$$\begin{aligned}\frac{\partial u}{\partial t} - 2\Omega_0 v + b_m u - \frac{\nu}{r}\frac{\partial f}{\partial \phi} &= -\frac{1}{\rho}\frac{\partial P'}{\partial r} + \omega^2 Be^{i(\sigma t - \phi)}\\\frac{\partial v}{\partial t} + 2\Omega_0 u + b_m v + \nu\frac{\partial f}{\partial r} &= -\frac{1}{\rho r}\frac{\partial P'}{\partial \phi} - i\omega^2 Be^{i(\sigma t - \phi)}\end{aligned} \tag{17}$$

$$P' = P - A\Omega_0^2 r\rho e^{-i\phi} \tag{18}$$

where $\sigma = \omega - \Omega_0$ is the frequency of oscillations of the free surface of the liquid (the velocity of wave propagation on the free surface of the liquid in the forward direction) and $\omega$ is the frequency of free oscillations (self-oscillations) of the system [35].

It is advisable to use special functions to solve Equation (17) taking into account Equation (7) and boundary conditions Equations (8)–(11).

Taking Equations (15)–(17) into account, we represent the velocity components of the liquid particle $u$ and $v$, as well as the pressure $P'$ and the function $f$ in the form:

$$G(r, \phi, t) = g(r)e^{i(\sigma t - \phi)} \tag{19}$$

where we consider forced oscillations of the fluid, as free fluctuations of the liquid quickly decay due to external friction and the viscosity of the liquid [36].

Taking Equation (19) into account, we can represent the function $f$ as:

$$f(r, \phi, t) = R(r)e^{i(\sigma t - \phi)} \tag{20}$$

After substituting Equation (20) into Equation (14), we will have the first-order Bessel equation with respect to $R(r)$, the solution to which is written as:

$$R(r) = z_1(\alpha r) = C_1 I_1(\alpha r) + C_2 N_1(\alpha r) \tag{21}$$

where

$$\alpha = \sqrt{-\frac{b_m + i\sigma}{\nu}} \tag{22}$$

where $I_1(\alpha r)$ and $N_1(\alpha r)$ are Bessel and Neumann functions of the first order; $C_1$ and $C_2$ are constants of integration, which are determined from the boundary conditions.

Using the continuity Equation (7) from relation Equation (6), taking into account Equations (20) and (21), we obtain the inhomogeneous Euler equation, solving which we obtain an expression for the component $u$ of the fluid particle velocity in the form:

$$u = \left[ C_3 + \frac{C_4}{r^2} + \frac{i}{\alpha^2 r} z_1(\alpha r) \right] e^{i(\sigma t - \phi)} \tag{23}$$

Taking Equation (23) into account, from the continuity equation, we find an expression for the component $v$ of the fluid particle velocity:

$$u = \left[ C_3 - \frac{C_4}{r^2} - \frac{i}{\alpha^2 r} z_1(\alpha r) + \frac{i z_0(\alpha r)}{\alpha} \right] e^{i(\sigma t - \phi)} \tag{24}$$

Now using Equations (19), (23) and (24) from the second equation of system Equation (17), we obtain an expression for $P'$:

$$P' = -i\rho r \left[ (2\Omega_0 + \sigma - ib_m)C_3 + \frac{2\Omega_0 - \sigma + ib_m}{r^2} C_4 - \frac{(2\Omega_0 - \sigma + ib_m)\nu}{(\sigma - ib_m)r} z_1(\alpha r) - \frac{\nu}{r} z_1(\alpha r) + iB\omega^2 \right] e^{i(\sigma t - \phi)} \tag{25}$$

The integration constants $C_1, C_2, C_3$ and $C_4$ are determined using the boundary conditions Equations (8)–(11). Taking into account Equations (18), (24) and (25) from Equations (2) and (3), we find the complex expression for the reaction force of a weakly conductive viscous fluid in the complex plane:

$$F_r = m_L \Omega_0^2 A e^{i\Omega_0 t} + m_L \omega^2 B \Phi e^{i\omega t} \tag{26}$$

where $F_r = F_x + iF_y$, $m_L = \pi \rho R^2 h$ is the mass of fluid required to completely fill the rotor cavity:

$$\Phi = \frac{\sigma \nu}{\Delta_0} \left\{ f_{12} \left[ \alpha I_0(\alpha R) - \frac{I_1(\alpha R)}{R} \right] - f_{11} \left[ \alpha N_0(\alpha R) - \frac{N_1(\alpha R)}{R} \right] + 1 \right\} \tag{27}$$

where

$$
\begin{aligned}
&\Delta_0 = f_{11}f_{14} - f_{12}f_{13} \\
&f_{11} = \frac{2R^2\alpha}{r_0^2} I_0(\alpha R) - \frac{4R}{r_0^2} I_1(\alpha R) + \left( \frac{4}{r_0} + \frac{i(\sigma - ib_m)}{\nu} r_0 \right) I_1(\alpha r_0) - 2\alpha I_0(\alpha r_0) \\
&f_{13} = \frac{\nu\alpha}{2(\sigma - ib_m)} \left[ \omega^2 - i\sigma b_m - q^2 \left( \Omega_0^2 - \sigma^2 + 2\Omega_0(\sigma - ib_m) + \frac{4\nu i\sigma}{r_0^2} \right) \right] I_0(\alpha R) + \\
&+ \frac{\nu R}{(\sigma - ib_m)r_0^2} \left( \Omega_0^2 + 2\Omega_0\sigma - \sigma^2 + i\sigma b_m + \frac{4\nu\sigma i}{r_0^2} \right) I_1(\alpha R) - \\
&- \frac{\nu}{(\sigma - ib_m)r_0} \left( \Omega_0^2 + 2\Omega_0\sigma + \frac{4\nu i\sigma}{r_0^2} \right) I_1(\alpha r_0) - \frac{2\nu i\sigma}{\alpha r_0^2} I_0(\alpha r_0)
\end{aligned}
\tag{28}
$$

Replacing the Bessel functions in the expressions for $f_{11}$ and $f_{13}$ by the Neumann functions of the same order, we obtain expressions for the functions $f_{12}$ and $f_{14}$. The quantity $q = \frac{R}{r_0}$ characterizes the degree of filling of the rotor cavity with liquid.

Now, when the fluid reaction force $F_r$ is known, we can determine the amplitude of forced oscillations and self-oscillations of the system [8,37–40].

Taking into account the identity of the first equation to the second, and the third to the fourth in system Equation (1), we will further consider a system of the form:

$$m\ddot{x} + 2c_0(x - x_2) + 2c_1(x - x_2)^3 + \chi\dot{x} = me\Omega_0^2 \cos\Omega_0 t + F_x$$
$$M\ddot{x}_2 + 2c_2 x_2 - 2c_0(x - x_2) - 2c_1(x - x_2)^3 + \chi_0\dot{x}_2 = 0 \tag{29}$$

Substituting the real parts from Equations (15), (16) and (26) into Equation (29), using the method of the imaging function, and making dimensionless each term of system Equation (29), we obtain a system of nonlinear algebraic equations for unknown dimensionless amplitudes $a$, $b$, $c$ and $d$. We obtain:

$$l_{11}a + l_{13}c = \mu s^2$$
$$l_{12}b + l_{14}d = 0$$
$$l_{13}c - (a - c) - \eta(a - c)\left[(a - c)^2 + 2(b - d)^2\right] = 0 \tag{30}$$
$$l_{14}d - (b - d) - \eta(b - d)\left[(b - d)^2 + 2(a - c)^2\right] = 0$$

where

$$l_{11} = -\mu_1 s^2 + ips; \; l_{12} = -(\mu + \mu_L\Phi)\tau^2 + ip\tau; \; l_{13} = (g_0 - s^2 + ip_0 s); \; l_{14} = (g_0 - \tau^2 + ip_0\tau).$$
$$n_0^2 = \frac{2c_0}{M}; \; n_1 = \frac{2c_1}{M}; \; n_2^2 = \frac{2c_2}{M}; \; \mu = \frac{m}{M}; \; \mu_L = \frac{m_L}{M}; \; m_L = \pi\rho R^2 H; \; \mu_1 = \mu + \mu_L; \; p = \frac{\chi}{Mn_0};$$
$$p_0 = \frac{\chi_0}{Mn_0}; \; g_0 = \frac{c_2}{c_0}; \; \eta = \frac{3\mu n_1 e^2}{4n_0^2}; \; a = \frac{A}{e}; \; b = \frac{B}{e};$$
$$c = \frac{C}{e}; \; d = \frac{D}{e} \tag{31}$$

where $\tau = \frac{\omega}{n_0}$ and $s = \frac{\Omega_0}{n_0}$ are the dimensionless frequency of natural vibrations of the system and the angular velocity of the rotor, respectively.

From the formulas that determine the unknowns $a$, $b$, $c$ and $d$, it is obvious that the amplitudes of the forced and natural oscillations of the system depend on the natural frequency of the system and they are interdependent [41]. This is one of the specific features of a nonlinear system. From formulas Equations (15) and (16), it follows that one more frequency is superimposed on the forced oscillations of the system, i.e., the system makes oscillations consisting of the sum of two harmonic oscillations [42]. In this case, the system performs a processional motion that differs from a circular precession [43].

From the first and second equations of system Equation (30), we have:

$$c = \beta_{13} + \beta_{11}a$$

$$a - c = a(1 - \beta_{11}) - \beta_{13} \tag{32}$$

$$d = -\frac{l_{12}}{l_{14}}b \tag{33}$$

$$b - d = b\left(1 + \frac{l_{12}}{l_{14}}\right) \tag{34}$$

$$b^2 = \lambda_2 a^2 + \lambda_1 a + \lambda_0 \tag{35}$$

where

$$\beta_{11} = -\frac{l_{11}}{l_{13}}; \; \beta_{13} = \frac{\mu s^2}{l_{13}}; \; \beta_{10} = -\frac{(l_{12}l_{14} + l_{12} + l_{14})}{\eta(l_{12} + l_{14})}; \; \lambda_0 = -\frac{2l_{14}^2(1 - \beta_{11})^2}{(l_{12} + l_{14})^2};$$
$$\lambda_1 = -\frac{4l_{14}^2(1 - \beta_{11})\beta_{13}}{(l_{12} + l_{14})^2}; \; \lambda_2 = (\beta_{10} - 2\beta_{13})\frac{l_{14}^2}{(l_{12} + l_{14})^2}$$

Now, from the third equation of system Equation (30), taking into account Equations (32), (33), and (35), we obtain an equation of the third power with respect to $a$:

$$a^3 + \delta_2 a^2 + \delta_1 a + \delta_0 = 0 \tag{36}$$

$$
\begin{aligned}
&\delta_2 = -\frac{3\mu s^2}{\beta_{12}}; \ \delta_1 = -\frac{l_{11}l_{13}^3 + \beta_{15}\mu s^2}{\beta_{12}\beta_{14}} + \frac{\beta_{16}}{\beta_{14}}; \ \delta_0 = \frac{\mu s^2\left(l_{13}^3 - \beta_{16}\right)}{\beta_{12}\beta_{14}}; \\
&\beta_{12} = l_{11} + l_{13}; \\
&\beta_{14} = 3\eta\beta_{12}^2; \ \beta_{15} = -6\eta\beta_{12}\mu s^2; \ \beta_{16} = 3\eta\mu^2 s^4 - 2\eta\beta_{10}l_{13}^2 - l_{13}^2
\end{aligned}
\tag{37}
$$

The cubic Equation (36) is solved analytically by the Cardano method and numerically by the Newton–Raphson method. The coefficients of Equation (36) are, in the general case, complex (when friction forces are taken into account). If friction forces are neglected, i.e., at $p = p_0 = 0$, the coefficients of Equation (36) become real [44]. In general, Equation (36) has three real roots. Taking into account Equation (37), from the solution of Equation (36), the unknown dimensionless amplitude of forced oscillations of the rotor $a$ is found, then from Equations (32), (33) and (35), the dimensionless amplitude of forced oscillations of the foundation $c$ is determined, as well as the dimensionless amplitudes of self-oscillations of the rotor $b$ and the foundation $d$. As can be seen from Equations (31)–(36), each value of the amplitude of the forced oscillations of the rotor $a$ corresponds to three values of the amplitude of the forced oscillations of the foundation $c$ and self-oscillations of the rotor and foundation $b$ and $d$.

Extreme values of the amplitude of forced oscillations depending on the angular velocity can be found from the following formulas:

$$\frac{\partial a}{\partial \Omega_0} = 0; \ \frac{\partial b}{\partial \Omega_0} = 0; \ \frac{\partial c}{\partial \Omega_0} = 0; \ \frac{\partial d}{\partial \Omega_0} = 0.$$

From the solution of this equation, we can find the values of the angular velocity of the rotor $\Omega_0$, at which the amplitudes of the forced and self-oscillations of the rotor and foundation will have maximum values [45].

From the obtained results, it is obvious that, in contrast to a linear system, in a nonlinear system, the amplitudes of natural oscillations (self-oscillations) of the rotor and foundation depend on the natural oscillation frequency of the system $\omega$.

Using smooth variations in $\omega$ at different values of $\Omega_0$ in the interval of changes in the rotor, foundation, liquid and magnetic field parameters (rotor dimensions, rotor mass $m$ and foundation mass $M$, coefficients of rigidity of rolling bearings and supports $c_0, c_1, c_2$ and external resistance coefficients $\chi, \chi_0$, the extent of filling of the rotor $q$, liquid viscosity $\nu$, magnetic field, etc.), it is possible to plot the dependences of the amplitude of forced oscillations and self-oscillations of the system, i.e., Amplitude–frequency characteristics of the rotor and the foundation [46].

By varying the frequencies of natural oscillations of the system $\omega$ in a wide range at the most required operating modes of the rotor speeds $\Omega_0$ (the angular speeds of rotor rotation required for the technological process) and other fixed parameters of the system using a PC, one can find the extreme values of the amplitudes of forced and self-oscillations of the rotor and the foundation (peaks of skeletal curves).

Based on the analysis of the results obtained, it is possible to find the optimal values of the system parameters, at which the amplitudes of forced oscillations and self-oscillations of the rotor and foundation will be minimal [47], i.e., so that they are significantly reduced over the entire range of the rotor system operation. In this case, the elastic foundation plays the role of a dynamic damper of rotor oscillations [48].

### 3. Special Cases

(1) In the case of an ideal fluid, i.e., for $v = 0$ and $b \neq 0$ from Equations (26) and (27), we obtain the hydrodynamic force and the function $\Phi$ in the form:

$$F_r = F_x + iF_y = Am_L\Omega_0^2 \exp(i\Omega_0 t) + Bm_L\omega^2\Phi \exp(i\omega t) \tag{38}$$

$$\Phi = \frac{\left(\sigma^2 - 2\Omega_0\sigma - \Omega_0^2 + i\gamma b_m\sigma\right)}{\left(\gamma\sigma^2 - 2\Omega_0\sigma - \Omega_0^2 + ib_m\sigma\right)} \tag{39}$$

or in the dimensionless form:

$$\frac{F_r}{eMn_0^2} = a\mu_L s^2 \exp(i\Omega_0 t) + b\mu_L\tau^2\Phi \exp(i\omega t)$$
$$\Phi = \frac{\left[\tau^2 - 4\tau s + s^2 + i\gamma b_n(\tau - s)\right]}{\left[\gamma\tau^2 - 2(\gamma+1)\tau s + (\gamma+1)s^2 + ib_n(\tau - s)\right]}.$$

where $\gamma = \frac{q^2+1}{q^2-1}$, $b_n = \frac{b_m}{n_0}$.

The case considered here, when the rotor cavity is partially filled with an ideal weakly conductive liquid, was chosen only for reasons of obtaining more or less simple formulas for an engineering assessment of the physical meaning of the process occurring in a nonlinear system [49].

(2) For synchronous precession of the rotor when $\omega = \Omega_0$, $\sigma = 0$, we have $\varphi = 1$, $F_r = (A + B)m_L\Omega_0^2 \exp(i\Omega_0 t)$, i.e., the foundation and the centrifuge perform circular synchronous precession. The main feature of synchronous precession is that the fluid is motionless with respect to the rotor cavity [13]. There is no wave motion of the fluid. In this case, the rotor and its foundation make forced movements caused by the unbalance of the rotor, and its cavity behaves as if it is completely filled with liquid. In this case, the magnetic field does not affect the motion of the system [50].

(3) For $v = 0$, $b_m = 0$ and $\tau \to s\left(1 + \frac{1}{\gamma}\left(1 \pm \sqrt{\gamma+1}\right)\right)$, the fluid reaction force tends to infinity, which will entail an unlimited increase in the amplitude of forced and natural oscillations of the rotor and its foundation [16].

(4) For $v = 0$, $b_m \to \infty$ and the reaction force of the liquid will take the form $F_r = Am_L\Omega_0^2 exp(i\Omega_0 t) + Bm_L\omega^2\gamma exp(i\omega t)$. In the case $\gamma = 1$ ((when the rotor cavity is completely filled with liquid), we obtain exactly the same picture as described in subparagraph 1. When $q \to \infty$ or $\gamma \to \infty$ (the amount of liquid in the cavity rapidly decreases), the amplitude of natural oscillations tends to infinity.

### 4. Free Oscillations of System

Let us consider a rotor system under the assumption that the rotor is balanced, with no imbalance, and solve the problem of oscillations of the rotor and the foundation [15], when the cavity of the first is partially filled with a weakly conductive viscous liquid. Using the equations of motion of system Equation (29) without imbalance, we obtain their solution in the form:

$$x = b_1 \cos\omega t \tag{40}$$

$$x_2 = d_1 \cos\omega t \tag{41}$$

Substituting Equations (40) and (41) into system Equation (29) and using the mapping function method, we obtain:

$$l_{12}b_1 + l_{14}d_1 = 0$$
$$l_{14}d_1 - (b_1 - d_1) - \eta_0(b_1 - d_1)^3 = 0 \tag{42}$$

where the coefficients of system Equation (42) have the same form, but the amplitudes of natural oscillations $b_1$ and $d_1$ have a linear dimension and the parameter $\eta = \eta_0 = \frac{3\mu n_1}{4n_0^2}$.

From the first system of Equation (42), we find:

$$d_1 = -\frac{l_{12}}{l_{14}} b_1 \qquad (43)$$

Substituting Equation (43) into the second equation of system Equation (42), we obtain expressions for the amplitude of natural oscillations of the rotor and foundation:

$$b_1 = \left[ -\frac{l_{14}^2 (l_{14}l_{12} + l_{12} + l_{14})}{\eta_0 (l_{12} + l_{14})^3} \right]^{-2} \qquad (44)$$

$$B = \left[ (\mathrm{Re}b_1)^2 + (\mathrm{Im}b_1)^2 \right]^{-2} \qquad (45)$$

$$D = \left[ (\mathrm{Re}d_1)^2 + (\mathrm{Im}d_1)^2 \right]^{-2} \qquad (46)$$

From the earlier and the last formulas [14], it is obvious that the amplitudes of natural oscillations of the rotor $B$ and foundation $D$ depend on the natural frequency $\omega$ of the nonlinear system [17–19]. For various fixed values of the angular speed of the rotor $\Omega_0$, by smoothly changing the value of the natural frequency $\omega$, it is possible to construct backbone curves, i.e., the dependence of the amplitudes $b_1$ and $d_1$ on $\omega$ and $\Omega_0$.

## 5. Results and Discussion

To evaluate the dependence of the damping and shift in frequencies of natural oscillations on the magnetic field, the operation of the rotor system with different values of parameters $b_m$ and $\gamma$ at the maximum amplitudes of the rotor system, which are observed at the main resonance, i.e., at $s = 1$, was considered (see Figures 2–11).

In the general case, three critical frequencies are observed in the system (see Figures 2 and 3). This means that due to the presence of liquid even in a small amount in the rotor cavity, two additional critical frequencies are superimposed with amplitudes, the values of which increase until the cavity is filled by one third [20]. A further increase in the amount of liquid in the cavity leads to the complete suppression of the second critical frequency, thus, at $\gamma = 1$ (complete filling of the cavity with liquid), two critical frequencies will be observed in the system.

An increase in the value of the parameter that characterizes the effect of a magnetic field on a liquid particle, in general, positively affects the dynamics of the system. When varying the $b_m$ parameter, a significant influence is observed for the amplitudes at the third critical frequency [21]. With an increase in $b_m$, the amplitudes of the second critical frequency are damped more weakly, and the amplitudes of the first critical frequency practically do not change, as they are also present in the absence of liquid in the rotor cavity. It should be noted that at a sufficiently high magnetic field strength, i.e., at $b_m = 5000$, the oscillations of the system are practically similar to the case of an empty rotor, as the liquid in this case, as if "solidifies" and behaves like a solid body [22]. No shift in critical frequencies is observed in all cases, except for $b_m = 5000$. For clarity and convenience of application of the results in engineering practice, the authors plotted the dependencies of critical amplitudes for different values of the parameters $b_m$ and $\gamma$ (see Figures 2 and 3).

With a small amount of liquid in the rotor cavity, for example, at $\gamma = 13.8$ ($r_0 = 0.93R$), three critical frequencies are observed at $\tau = 0.16$, $\tau = 0.78$, and $\tau = 1.36$, and three zones of self-oscillations, the maximum amplitudes of the rotor and foundation are observed at the third critical speed in cases $b_m = 0.100$. Due to the small amount of liquid, the second critical frequency has smaller amplitudes compared to the first. With an increase in the $b_m$ parameter to 500, a slight shift in the second and third critical frequencies is observed towards an increase in the angular velocities of the rotor rotation [23,24]. In the case of a sufficiently high magnetic field strength, the second critical frequency shifts in the direction of increasing dimensionless frequency up to $\tau = 1$, whereas the third critical

frequency of the rotor and foundation is completely damped. The numerical values of the amplitudes for this case are presented in Table 1. Dashes in the table indicate the absence of critical frequencies or the complete damping of the natural oscillation amplitudes in this interval [25].

**Table 1.** Critical Speeds.

| $B, \gamma = 13.8$ | First Critical Speed Amplitude Value | Second Critical Speed Amplitude Value | Third Critical Speed Amplitude Value |
|---|---|---|---|
| $b_m = 0$ | 0.1068045741, $\tau = 0.16$ | 0.090720226, $\tau = 0.78$ | 0.345717672, $\tau = 1.36$ |
| $b_m = 0.5$ | 0.106796702, $\tau = 0.16$ | 0.090718, $\tau = 0.78$ | 0.345709, $\tau = 1.36$ |
| $b_m = 5$ | 0.106725474, $\tau = 0.16$ | 0.09065, $\tau = 0.78$ | 0.341007, $\tau = 1.36$ |
| $b_m = 25$ | 0.10640083, $\tau = 0.16$ | 0.089259, $\tau = 0.78$ | 0.256448, $\tau = 1.36$ |
| $b_m = 50$ | 0.105976946, $\tau = 0.16$ | 0.085366, $\tau = 0.78$ | 0.155548, $\tau = 1.36$ |
| $b_m = 100$ | 0.105071989, $\tau = 0.16$ | 0.073742909, $\tau = 0.78$ | 0.109144682, $\tau = 1.36$ |
| $b_m = 500$ | 0.095853985, $\tau = 0.16$ | 0.039189072, $\tau = 0.79$ | 0.03976382, $\tau = 1.43$ |
| $b_m = 5000$ | 0.054819732, $\tau = 0.15$ | 0.032983197, $\tau = 1$ | – |
| $D, \gamma = 13.8$ | First Critical Speed Amplitude Value | Second Critical Speed Amplitude Value | Third Critical Speed Amplitude Value |
| $b_m = 0$ | 0.066385927, $\tau = 0.16$ | 0.040110122, $\tau = 0.78$ | 0.453737506, $\tau = 1.36$ |
| $b_m = 0.5$ | 0.066382727, $\tau = 0.16$ | 0.040109872, $\tau = 0.78$ | 0.45372321, $\tau = 1.36$ |
| $b_m = 5$ | 0.066353881, $\tau = 0.16$ | 0.040087202, $\tau = 0.78$ | 0.447400839, $\tau = 1.36$ |
| $b_m = 25$ | 0.066224805, $\tau = 0.16$ | 0.045069061, $\tau = 0.79$ | 0.334005995, $\tau = 1.36$ |
| $b_m = 50$ | 0.066061617, $\tau = 0.16$ | 0.048328381, $\tau = 0.79$ | 0.198268343, $\tau = 1.36$ |
| $b_m = 100$ | 0.065730061, $\tau = 0.16$ | 0.041947113, $\tau = 0.79$ | 0.088761642, $\tau = 1.36$ |
| $b_m = 500$ | 0.063037959, $\tau = 0.16$ | 0.01798351, $\tau = 0.79$ | 0.016219871, $\tau = 1.37$ |
| $b_m = 5000$ | 0.054819732, $\tau = 0.16$ | 0.014450226, $\tau = 0.85$ | – |

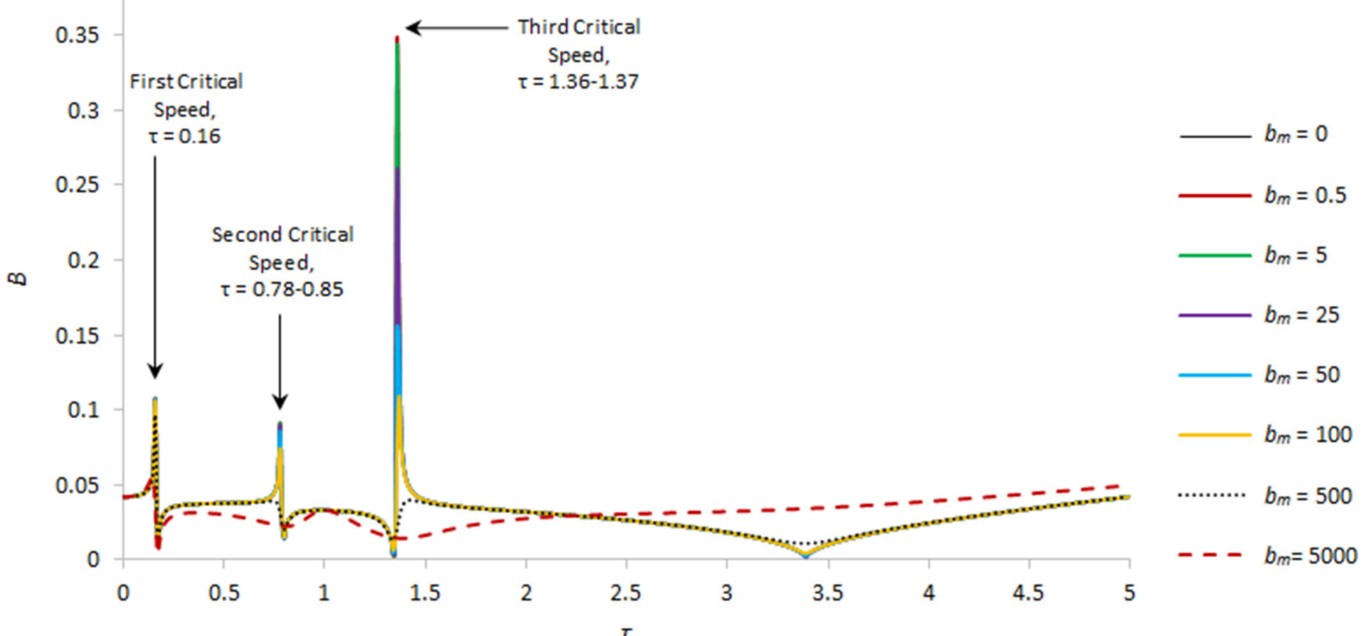

**Figure 2.** Amplitude–frequency characteristics of natural oscillations of the rotor for *s* = 1 and $\gamma$ = 13.8.

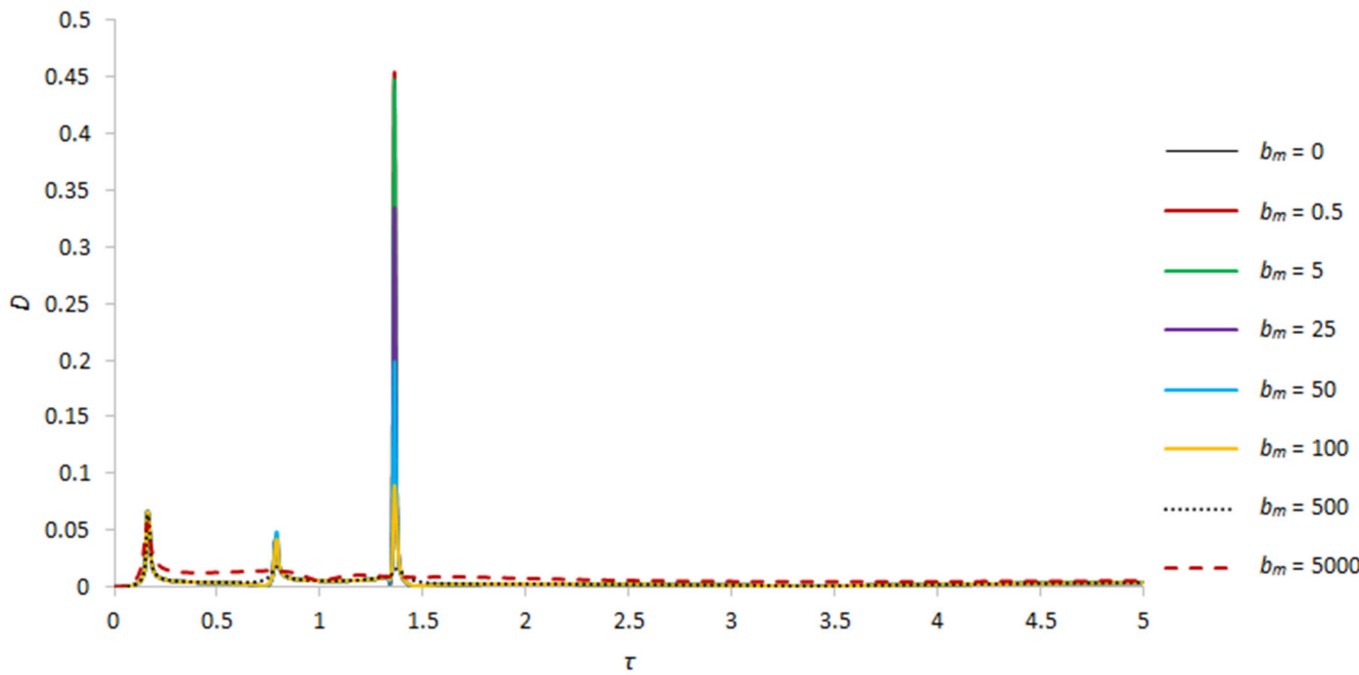

**Figure 3.** Amplitude–frequency characteristics of natural oscillations of the foundation for $s = 1$ and $\gamma = 13.8$.

As the extent of filling increases, as mentioned above, the amplitudes of the second and third critical frequencies increase (see Figures 4 and 5). For example, at $\gamma = 4.56$ ($r_0 = 0.8R$) and $b_m = 0$, the maximum values of the rotor amplitudes at the second and third critical frequencies are 3.3-fold and 5.4-fold greater than the values for the same case at $\gamma = 13.8$, and are 2.64-fold and 17.45-fold greater than the amplitudes at the first critical frequency, i.e., at $\tau = 0.16$. Moreover, an increase in the amount of liquid in the rotor cavity affects the displacement of the second and third critical frequencies [26]. For example, in this case, the second critical frequency occurs a little earlier, compared with the case of $\gamma = 13.8$, at $\tau = 0.69$, whereas the third critical frequency appears a little later, at $\tau = 1.76$. With an increase in the parameter $b_m$, for example, at $b_m = 500$ and $b_m = 5000$, the oscillation amplitudes at the second and third critical speeds of the rotor and foundation, as expected, are damped up to their complete damping [27], which also indicates a positive effect of the high-frequency magnetic field on the system. The numerical values of the amplitudes for this case are presented in Table 2.

**Table 2.** Critical Speeds.

| $B, \gamma = 4.56$ | First Critical Speed Amplitude Value | Second Critical Speed Amplitude Value | Third Critical Speed Amplitude Value |
|---|---|---|---|
| $b_m = 0$ | 0.110805396, $\tau = 0.16$ | 0.292896271, $\tau = 0.69$ | 1.917205238, $\tau = 1.76$ |
| $b_m = 0.5$ | 0.110797785, $\tau = 0.16$ | 0.292762234, $\tau = 0.69$ | 1.915895568, $\tau = 1.76$ |
| $b_m = 5$ | 0.110729214, $\tau = 0.16$ | 0.286890823, $\tau = 0.69$ | 1.710435466, $\tau = 1.76$ |
| $b_m = 25$ | 0.110422857, $\tau = 0.16$ | 0.205546804, $\tau = 0.69$ | 0.558152699, $\tau = 1.76$ |
| $b_m = 50$ | 0.110036396, $\tau = 0.16$ | 0.124228936, $\tau = 0.69$ | 0.231023687, $\tau = 1.76$ |
| $b_m = 100$ | 0.10925273, $\tau = 0.16$ | 0.066849372, $\tau = 0.69$ | 0.104944945, $\tau = 1.76$ |
| $b_m = 500$ | 0.102684806, $\tau = 0.16$ | – | – |
| $b_m = 5000$ | 0.08448214, $\tau = 0.15$ | – | – |

**Table 2.** *Cont.*

| $D$, $\gamma = 4.56$ | First Critical Speed Amplitude Value | Second Critical Speed Amplitude Value | Third Critical Speed Amplitude Value |
|---|---|---|---|
| $b_m = 0$ | 0.07047227, $\tau = 0.16$ | 0.226388549, $\tau = 0.69$ | 1.704994351, $\tau = 1.76$ |
| $b_m = 0.5$ | 0.070468917, $\tau = 0.16$ | 0.226287308, $\tau = 0.69$ | 1.70381006, $\tau = 1.76$ |
| $b_m = 5$ | 0.07043893, $\tau = 0.16$ | 0.221699183, $\tau = 0.69$ | 1.520112352, $\tau = 1.76$ |
| $b_m = 25$ | 0.070309814, $\tau = 0.16$ | 0.157597686, $\tau = 0.69$ | 0.488944743, $\tau = 1.76$ |
| $b_m = 50$ | 0.070157935, $\tau = 0.16$ | 0.034896767, $\tau = 0.7$ | 0.194103909, $\tau = 1.76$ |
| $b_m = 100$ | 0.069885617, $\tau = 0.16$ | 0.045941501, $\tau = 0.69$ | 0.072065559, $\tau = 1.76$ |
| $b_m = 500$ | 0.069093807, $\tau = 0.16$ | 0.012824952, $\tau = 0.7$ | 0.010085603, $\tau = 1.77$ |
| $b_m = 5000$ | 0.089252733, $\tau = 0.16$ | – | – |

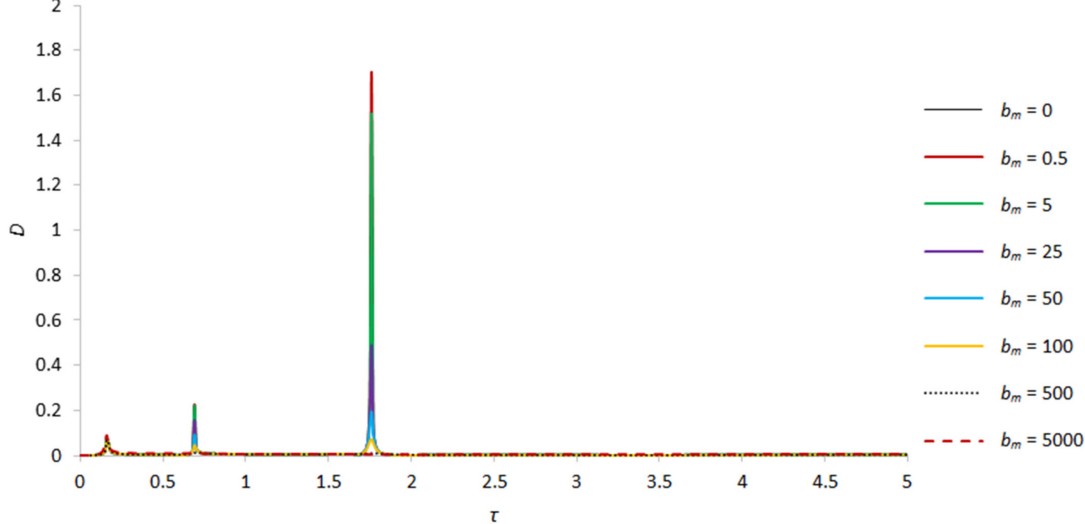

**Figure 4.** Amplitude–frequency characteristics of natural oscillations of the rotor for $\gamma = 4.56$, $s = 1$.

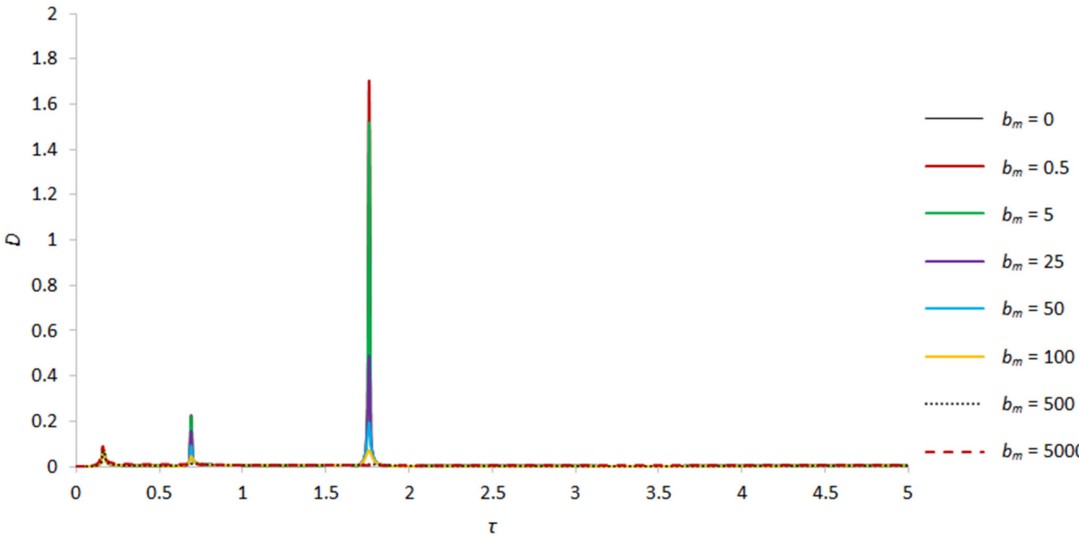

**Figure 5.** Amplitude–frequency characteristics of natural oscillations of the foundation for $\gamma = 4.56$, $s = 1$.

When the rotor cavity is filled with liquid by one third, i.e., at $\gamma = 2.6$, the system still has three critical frequencies ($\tau = 0.16$, $\tau = 0.64$ and $\tau = 2.14$, see Figures 6 and 7). In this case, the maximum values of the rotor amplitudes at the third critical frequency are 19.8-fold greater than the amplitudes in the similar case at $\gamma = 4.56$ ($b_m = 0$), whereas the amplitudes

of the second critical frequency are comparable with the previous case. An increase in the parameter that determines the influence of the magnetic field on a fluid particle from 0 to 0.5 leads to damping of the rotor and foundation amplitudes corresponding to the third critical frequency [28]. A significant decrease in the amplitudes of the second critical frequency occurs with an increase in $b_m$ to 25. Further, as $b_m$ increases to 500, the amplitudes of the second and third critical frequencies are almost completely suppressed. A change in the parameter that determines the influence of the magnetic field on a liquid particle has practically no effect on the amplitudes of the first critical frequency, except for large values of the magnetic field strength. It should be noted that in this case the third critical frequency shifts quite strongly to the right, in the direction of increasing natural frequencies, from 1.76 to 2.14, which also imposes certain restrictions on the choice of operating speed range in the absence of magnetic field influence [29]. The numerical values of the amplitudes for this case are presented in Table 3.

**Table 3.** Critical Speeds.

| $B$, $\gamma$ = 2.6 | First Critical Speed Amplitude Value | Second Critical Speed Amplitude Value | Third Critical Speed Amplitude Value |
|---|---|---|---|
| $b_m = 0$ | | 0.188180917, $\tau$ = 0.64 | 38.0983794, $\tau$ = 2.14 |
| $b_m = 0.5$ | | 0.188128512, $\tau$ = 0.64 | 35.05368501, $\tau$ = 2.14 |
| $b_m = 5$ | 0.114550189, $\tau$ = 0.16 | 0.185743424, $\tau$ = 0.64 | 4.620198303, $\tau$ = 2.14 |
| $b_m = 25$ | 0.114543039, $\tau$ = 0.16 | 0.148131171, $\tau$ = 0.64 | 0.442357104, $\tau$ = 2.14 |
| $b_m = 50$ | 0.113852547, $\tau$ = 0.16 | 0.100918717, $\tau$ = 0.64 | 0.180448011, $\tau$ = 2.15 |
| $b_m = 100$ | 0.113189756, $\tau$ = 0.16 | 0.06088884, $\tau$ = 0.64 | 0.093096022, $\tau$ = 2.16 |
| $b_m = 500$ | 0.109089699, $\tau$ = 0.16 | – | – |
| $b_m = 5000$ | 0.124710542, $\tau$ = 0.16 | – | – |
| $D$, $\gamma$ = 2.6 | First Critical Speed Amplitude Value | Second Critical Speed Amplitude Value | Third Critical Speed Amplitude Value |
| $b_m = 0$ | 0.074299992, $\tau$ = 0.16 | 0.130906778, $\tau$ = 0.64 | 37.42981166, $\tau$ = 2.14 |
| $b_m = 0.5$ | 0.074296638, $\tau$ = 0.16 | 0.130872119, $\tau$ = 0.64 | 34.43775859, $\tau$ = 2.14 |
| $b_m = 5$ | 0.074266809, $\tau$ = 0.16 | 0.129169584, $\tau$ = 0.64 | 4.531230535, $\tau$ = 2.14 |
| $b_m = 25$ | 0.074142119, $\tau$ = 0.16 | 0.101854993, $\tau$ = 0.64 | 0.417323892, $\tau$ = 2.14 |
| $b_m = 50$ | 0.074004213, $\tau$ = 0.16 | 0.067028512, $\tau$ = 0.64 | 0.148186179, $\tau$ = 2.14 |
| $b_m = 100$ | 0.073787317, $\tau$ = 0.16 | 0.036021818, $\tau$ = 0.64 | 0.053425977, $\tau$ = 2.14 |
| $b_m = 500$ | 0.074595252, $\tau$ = 0.16 | 0.01092513, $\tau$ = 0.66 | 0.007393946, $\tau$ = 2.14 |
| $b_m = 5000$ | 0.140446071, $\tau$ = 0.16 | – | – |

When the rotor cavity is half and two-thirds full, i.e., at $\gamma$ = 1.67 and $\gamma$ = 1.25, the character of changes in the amplitude and frequency of natural oscillations of the system is similar to the case when the rotor cavity is one-third filled. In this case, the system oscillates with smaller amplitudes by almost an order of magnitude lower than when the rotor cavity is one-third full (see Figures 8 and 9). In the case of half filling, the second and third critical frequencies are still greater than the first critical frequency [51]. Accordingly, a stronger shift in critical frequencies is observed compared to those listed above, for example, the second critical frequency appears already at $\tau$ = 0.61, while the third already at $\tau$ = 2.61.

When the cavity is filled by two-thirds, the amplitudes of the first and second critical frequencies are commensurate with each other and with the above case. The second critical frequency shifts in the direction of decreasing angular velocities and appears at $\tau$ = 0.59, whereas the third critical frequency shifts to the left in the direction of its increase and appears at $\tau$ = 3.02. With an increase in the parameter $b_m$, the oscillation amplitudes at the second and third critical speeds of the rotor and foundation, as before, are damped until their full damping. The numerical values of the amplitudes for this case are presented in Tables 4 and 5.

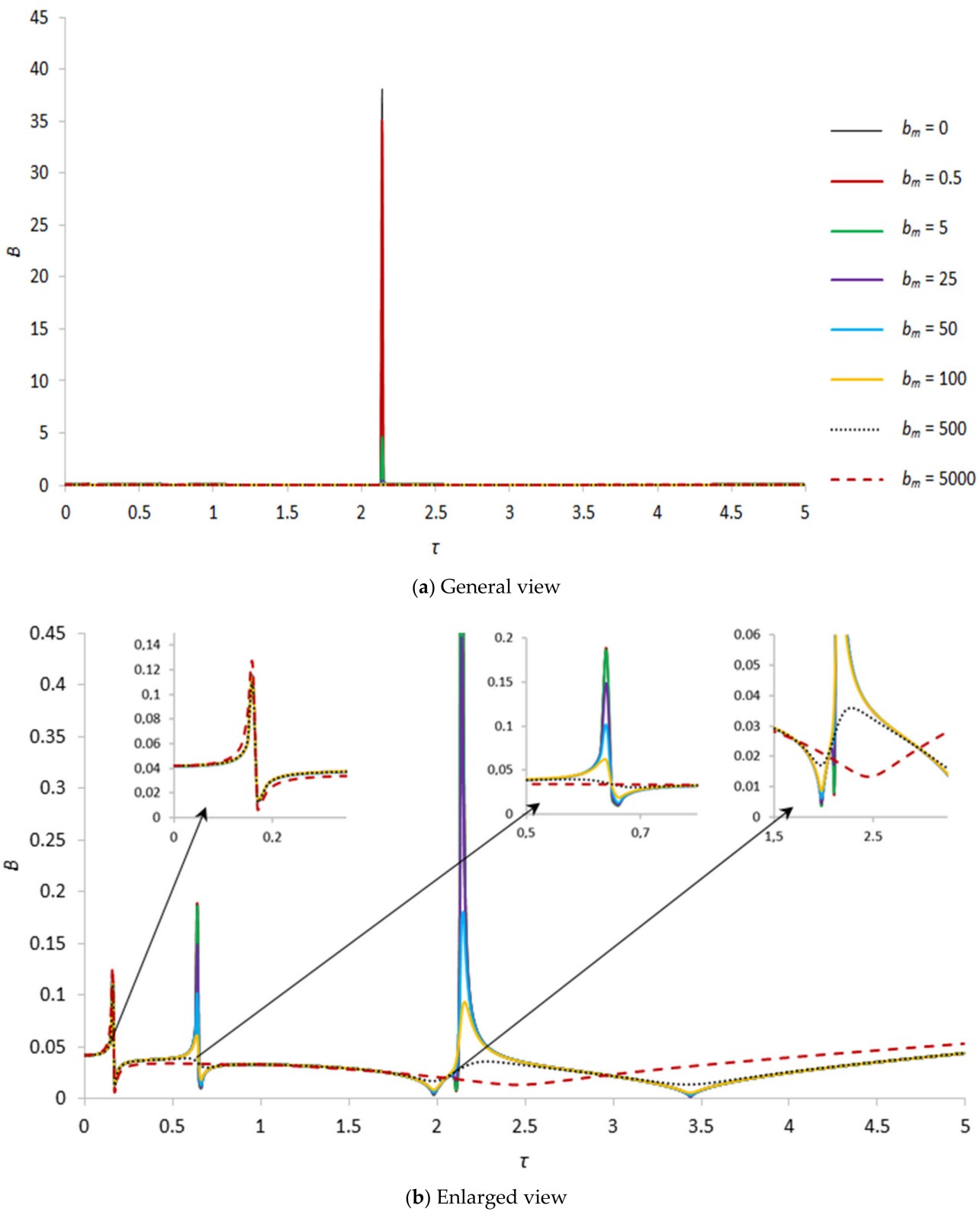

**Figure 6.** Amplitude–frequency characteristic of natural oscillations of the rotor for $s = 1$ and $\gamma = 2.6$; (**a**) general view; (**b**) enlarged view.

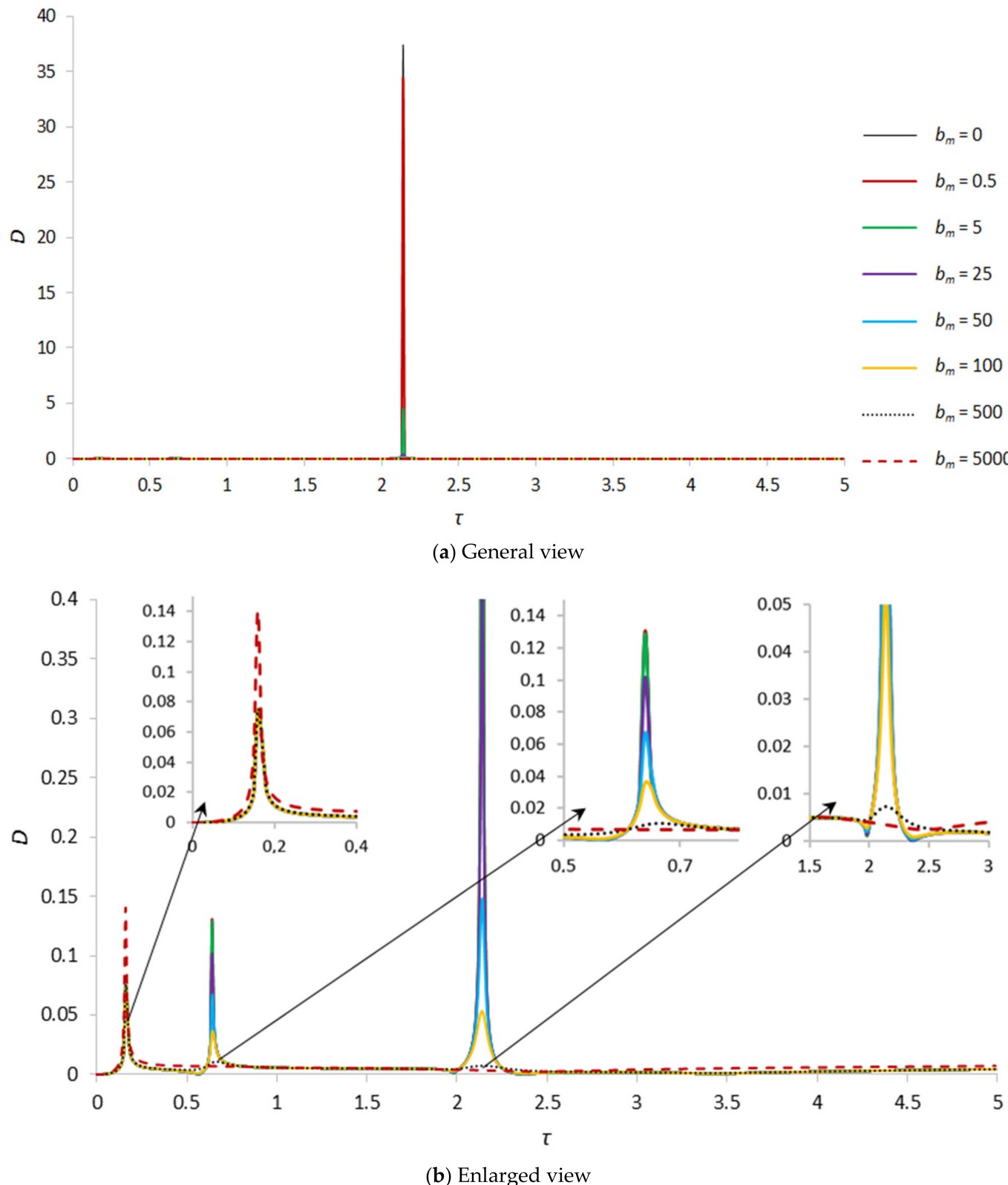

(**a**) General view

(**b**) Enlarged view

**Figure 7.** Amplitude–frequency characteristic of natural oscillations of the foundation for $s = 1$ and $\gamma = 2.6$; (**a**) general view; (**b**) enlarged view.

**Table 4.** Critical Speeds.

| $B, \gamma = 1.67$ | First Critical Speed Amplitude Value | Second Critical Speed Amplitude Value | Third Critical Speed Amplitude Value |
|---|---|---|---|
| $b_m = 0$ | 0.118574461, $\tau = 0.16$ | 0.27197385, $\tau = 0.61$ | 2.395991343, $\tau = 2.61$ |
| $b_m = 0.5$ | 0.11856806, $\tau = 0.16$ | 0.271326026, $\tau = 0.61$ | 2.386092738, $\tau = 2.61$ |
| $b_m = 5$ | 0.118510785, $\tau = 0.16$ | 0.244571454, $\tau = 0.61$ | 1.657566825, $\tau = 2.61$ |
| $b_m = 25$ | 0.118263621, $\tau = 0.16$ | 0.108761599, $\tau = 0.61$ | 0.307975554, $\tau = 2.61$ |
| $b_m = 50$ | 0.117971542, $\tau = 0.16$ | 0.079987206, $\tau = 0.61$ | 0.141437807, $\tau = 2.61$ |
| $b_m = 100$ | 0.117443017, $\tau = 0.16$ | 0.058212949, $\tau = 0.6$ | 0.078056973, $\tau = 2.61$ |
| $b_m = 500$ | 0.115713077, $\tau = 0.16$ | – | – |
| $b_m = 5000$ | 0.200140464, $\tau = 0.16$ | – | – |
| $D, \gamma = 1.67$ | First Critical Speed Amplitude Value | Second Critical Speed Amplitude Value | Third Critical Speed Amplitude Value |
| $b_m = 0$ | 0.078416443, $\tau = 0.16$ | 0.319661082, $\tau = 0.61$ | 2.101480075, $\tau = 2.61$ |
| $b_m = 0.5$ | 0.078413261, $\tau = 0.16$ | 0.318899815, $\tau = 0.61$ | 2.092707266, $\tau = 2.61$ |
| $b_m = 5$ | 0.078385116, $\tau = 0.16$ | 0.286969287, $\tau = 0.61$ | 1.448573522, $\tau = 2.61$ |
| $b_m = 25$ | 0.078270549, $\tau = 0.16$ | 0.122332756, $\tau = 0.61$ | 0.248941102, $\tau = 2.61$ |
| $b_m = 50$ | 0.078151336, $\tau = 0.16$ | 0.064906783, $\tau = 0.61$ | 0.091711735, $\tau = 2.61$ |
| $b_m = 100$ | 0.077991713, $\tau = 0.16$ | 0.03343016, $\tau = 0.6$ | 0.034567932, $\tau = 2.61$ |
| $b_m = 500$ | 0.080131667, $\tau = 0.16$ | – | – |
| $b_m = 5000$ | 0.193818388, $\tau = 0.16$ | – | – |

**Table 5.** Critical Speeds.

| $B, \gamma = 1.25$ | First Critical Speed Amplitude Value | Second Critical Speed Amplitude Value | Third Critical Speed Amplitude Value |
|---|---|---|---|
| $b_m = 0$ | 0.121638282, $\tau = 0.16$ | 0.122951082, $\tau = 0.59$ | 2.281429046, $\tau = 3.02$ |
| $b_m = 0.5$ | 0.121632639, $\tau = 0.16$ | 0.122901957, $\tau = 0.59$ | 2.25145796, $\tau = 3.02$ |
| $b_m = 5$ | 0.121582276, $\tau = 0.16$ | 0.121641647, $\tau = 0.59$ | 1.035804071, $\tau = 3.02$ |
| $b_m = 25$ | 0.121367604, $\tau = 0.16$ | 0.098499341, $\tau = 0.58$ | 0.166361948, $\tau = 3.02$ |
| $b_m = 50$ | 0.121120227, $\tau = 0.16$ | 0.078839058, $\tau = 0.58$ | 0.103151902, $\tau = 3.02$ |
| $b_m = 100$ | 0.120694759, $\tau = 0.16$ | 0.055532351, $\tau = 0.58$ | 0.06308127, $\tau = 3.02$ |
| $b_m = 500$ | 0.120406103, $\tau = 0.16$ | – | – |
| $b_m = 5000$ | 0.210144507, $\tau = 0.16$ | – | – |
| $D, \gamma = 1.25$ | First Critical Speed Amplitude Value | Second Critical Speed Amplitude Value | Third Critical Speed Amplitude Value |
| $b_m = 0$ | 0.081552542, $\tau = 0.16$ | 0.154403646, $\tau = 0.59$ | 2.6271788, $\tau = 3.02$ |
| $b_m = 0.5$ | 0.081549629, $\tau = 0.16$ | 0.154341266, $\tau = 0.59$ | 2.59220114, $\tau = 3.02$ |
| $b_m = 5$ | 0.081523936, $\tau = 0.16$ | 0.152395469, $\tau = 0.59$ | 1.173065248, $\tau = 3.02$ |
| $b_m = 25$ | 0.081421162, $\tau = 0.16$ | 0.098951316, $\tau = 0.58$ | 0.14148849, $\tau = 3.02$ |
| $b_m = 50$ | 0.081318754, $\tau = 0.16$ | 0.03067025, $\tau = 0.59$ | 0.051515196, $\tau = 3.02$ |
| $b_m = 100$ | 0.081199692, $\tau = 0.16$ | 0.055532351, $\tau = 0.59$ | 0.020073614, $\tau = 3.02$ |
| $b_m = 500$ | 0.083967196, $\tau = 0.16$ | – | – |
| $b_m = 5000$ | 0.186578729, $\tau = 0.16$ | – | – |

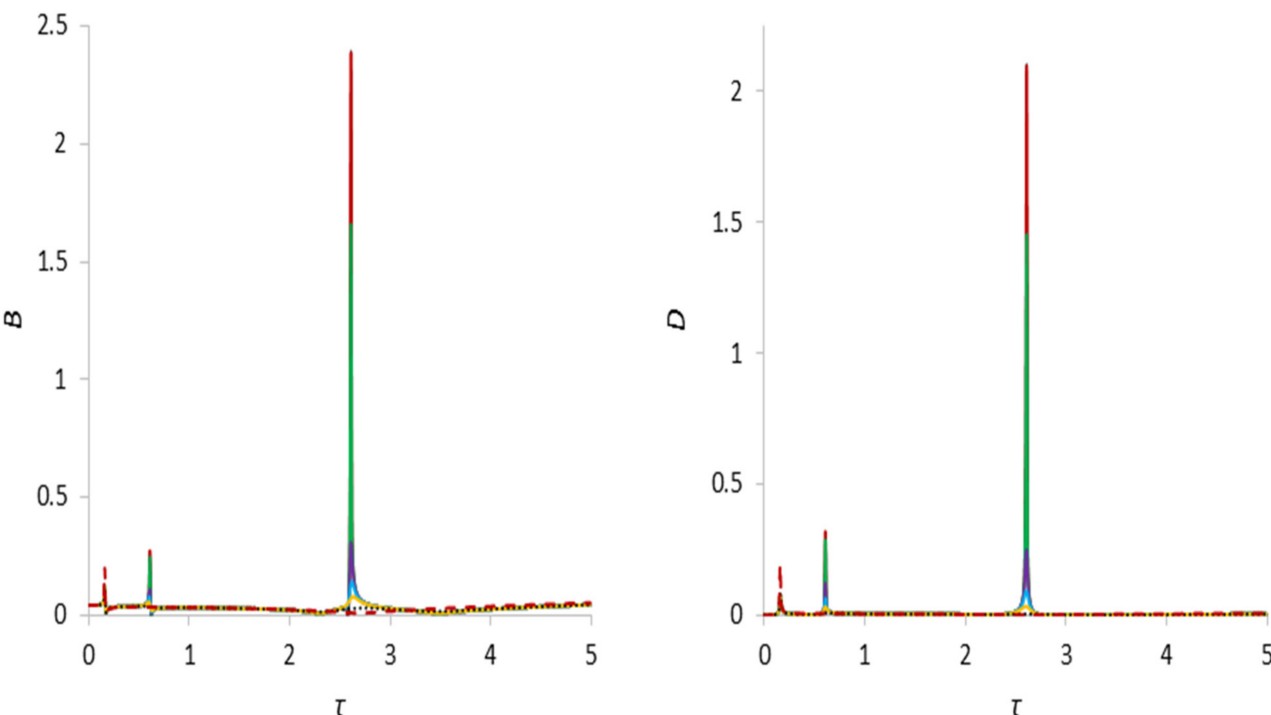

**Figure 8.** Amplitude–frequency characteristic of natural oscillations of the rotor and foundation for $s = 1$ and $\gamma = 1.67$.

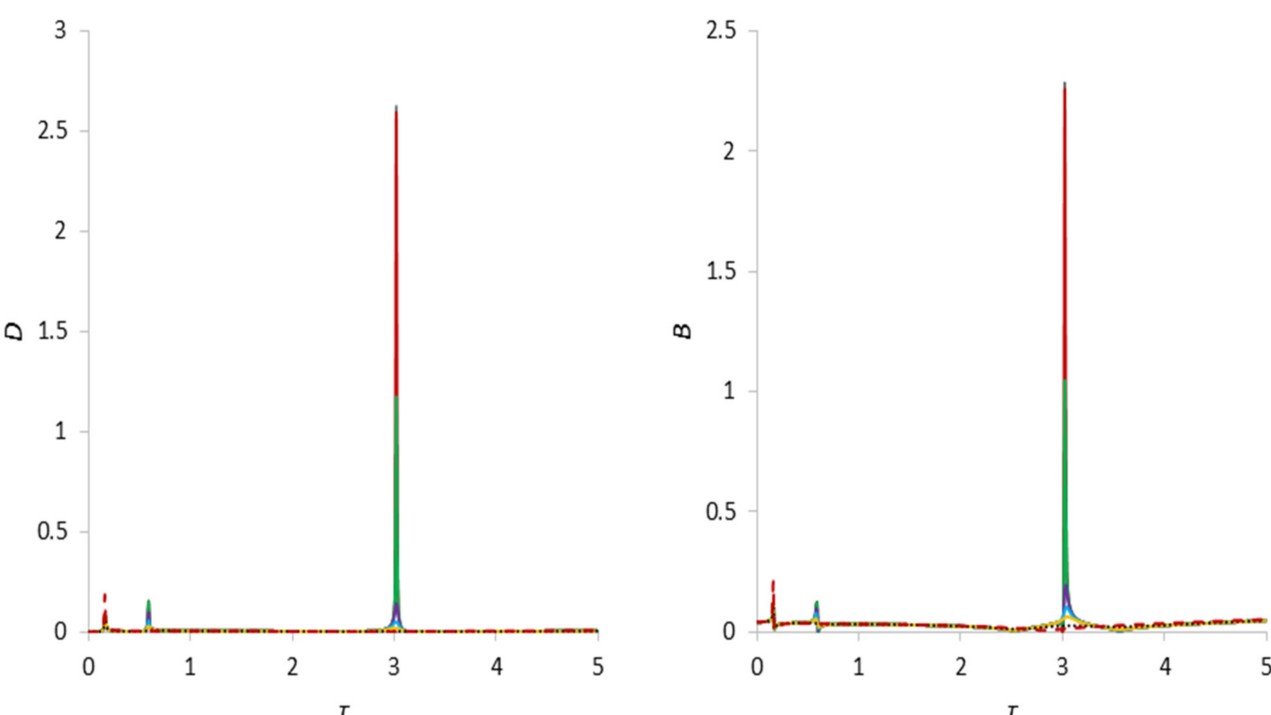

**Figure 9.** Amplitude–frequency characteristic of natural oscillations of the rotor and foundation for $s = 1$ and $\gamma = 1.25$.

When the rotor cavity is almost completely filled with liquid, for example, at $\gamma = 1.03$ ($r_0 = 0.125R$), the amplitudes at the second critical frequency are smaller than the amplitudes at the first critical frequency, whereas the amplitudes at the third critical frequency are several-fold smaller than the values observed in the case when the cavity is filled with liquid by half and by two thirds (see Figures 10 and 11). The shift in critical frequencies due

to an increase in the amount of liquid in the cavity in this case is even more pronounced. For example, the second critical frequency in this case appears already at $\tau = 0.57$ and the third already at $\tau = 3.36$. This means that with an increase in filling, the second critical frequency shifts to the first critical frequency until it merges with it, whereas the third critical frequency shifts in the direction of increasing frequency to the right and is almost completely damped with an increase in the amount of fluid in the rotor cavity. As before, with an increase in the parameter characterizing the influence of the magnetic field on a liquid particle $b_m$, the amplitudes of the second and third critical frequencies are damped up to their complete dampening. The numerical values of the amplitudes for this case are presented in Table 6.

**Table 6.** Critical Speeds.

| $B, \gamma = 1.03$ | First Critical Speed Amplitude Value | Second Critical Speed Amplitude Value | Third Critical Speed Amplitude Value |
|---|---|---|---|
| $b_m = 0$ | $0.123836133, \tau = 0.16$ | $0.154082698, \tau = 0.57$ | $0.582653298, \tau = 3.37$ |
| $b_m = 0.5$ | $0.123831142, \tau = 0.16$ | $0.15403417, \tau = 0.57$ | $0.580389919, \tau = 3.37$ |
| $b_m = 5$ | $0.123786664, \tau = 0.16$ | $0.151690752, \tau = 0.57$ | $0.425847453, \tau = 3.37$ |
| $b_m = 25$ | $0.123598669, \tau = 0.16$ | $0.117230408, \tau = 0.57$ | $0.114353556, \tau = 3.37$ |
| $b_m = 50$ | $0.123385864, \tau = 0.16$ | $0.080130589, \tau = 0.57$ | $0.069492128, \tau = 3.38$ |
| $b_m = 100$ | $0.123033788, \tau = 0.16$ | $0.054080999, \tau = 0.56$ | $0.046679793, \tau = 3.37$ |
| $b_m = 500$ | $0.123518243, \tau = 0.16$ | – | – |
| $b_m = 5000$ | $0.196425395, \tau = 0.16$ | – | – |

| $D, \gamma = 1.03$ | First Critical Speed Amplitude Value | Second Critical Speed Amplitude Value | Third Critical Speed Amplitude Value |
|---|---|---|---|
| $b_m = 0$ | $0.083803356, \tau = 0.16$ | $0.101565041, \tau = 0.57$ | $0.124078552, \tau = 3.35$ |
| $b_m = 0.5$ | $0.083800714, \tau = 0.16$ | $0.101535013, \tau = 0.57$ | $0.123990942, \tau = 3.35$ |
| $b_m = 5$ | $0.08377746, \tau = 0.16$ | $0.099920685, \tau = 0.57$ | $0.115971317, \tau = 3.35$ |
| $b_m = 25$ | $0.083685484, \tau = 0.16$ | $0.075582465, \tau = 0.57$ | $0.050630183, \tau = 3.35$ |
| $b_m = 50$ | $0.08359651, \tau = 0.16$ | $0.048524802, \tau = 0.57$ | $0.022746045, \tau = 3.35$ |
| $b_m = 100$ | $0.083504258, \tau = 0.16$ | $0.028624914, \tau = 0.58$ | $0.010481636, \tau = 3.33$ |
| $b_m = 500$ | $0.08647026, \tau = 0.16$ | $0.009075681, \tau = 0.62$ | – |
| $b_m = 5000$ | $0.165670988, \tau = 0.16$ | – | – |

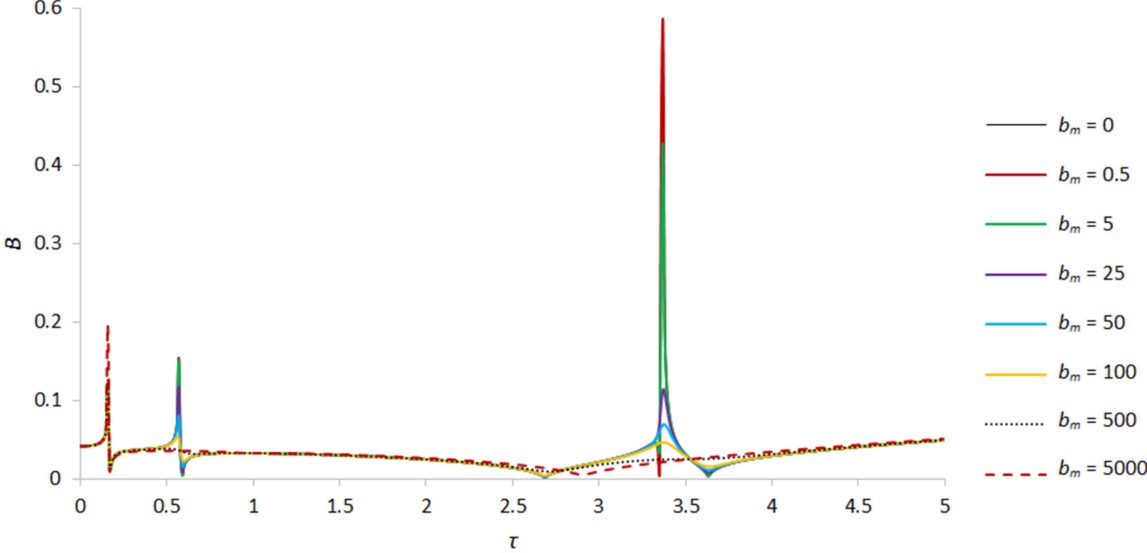

**Figure 10.** Amplitude–frequency characteristic of natural oscillations of the rotor for $s = 1$ and $\gamma = 1.03$.

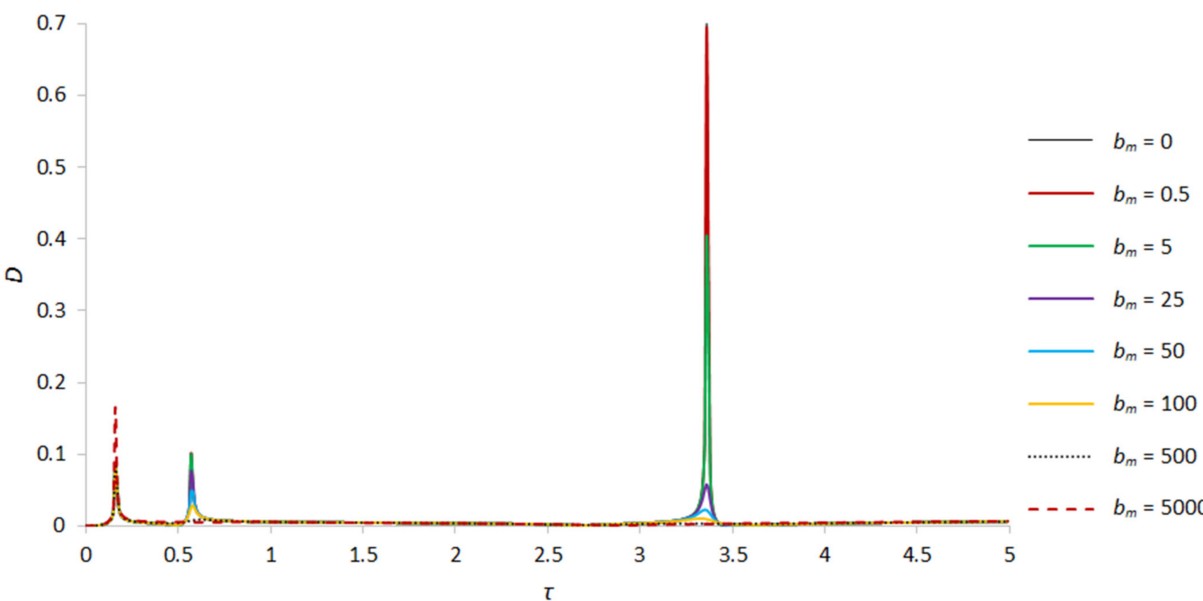

**Figure 11.** Amplitude–frequency characteristic of natural oscillations of the foundation for *s* = 1 and $\gamma$ = 1.03.

The electrical conductivity of the liquid is directly proportional to the length of the liquid volume in the rotor cavity and inversely proportional to the cross-sectional area of this volume and its electrical resistance. The intensity of a uniform axial magnetic field is proportional to the current strength, length and number of turns of the solenoid. Thus, due to sufficient damping of the amplitudes of the second and third critical frequencies, as well as due to its simpler implementation from the technical point of view, for further studying of the effect of a high-frequency magnetic field on nonlinear oscillations it was decided to use the value of the parameter $b_m = 100$. The shift in critical frequencies depends not only on the amount of liquid being filled but also on the frequency of the disturbing force, in our case, on the angular velocity of the rotor. To assess the shift in critical frequencies, diagrams were constructed where the abscissa axes correspond to the dimensionless angular velocity of rotation of the rotor *s* and the ordinate axes to the dimensionless natural frequency $\tau$.

## 6. Conclusions

The analysis of sources in this field of research shows that this work is the first where an analytically generalized dynamic model of the "rotor–weakly conductive fluid–foundation" system, which takes into account the electromagnetic properties of a viscous fluid and its vibrations, the nonlinear stiffness properties of bearing supports and vibrations of the foundation, was developed and solved. Due to the presence of liquid in the rotor cavity, several additional critical frequencies are imposed on the system, whose amplitudes in some cases exceed the amplitudes of the empty rotor by several orders of magnitude, which also imposes certain restrictions on the choice of operating speeds. An increase in the value of the parameter that characterizes the influence of the magnetic field on a fluid particle leads to damping of the amplitudes of the natural oscillations of the rotor and the foundation, which generally has a positive effect on the dynamics of the system. In this case, a change in the value of the magnetic field strength has practically no effect on the shifts in critical frequencies. In case of synchronous precession, the fluid is motionless with respect to the rotor cavity. There is no wave motion of the liquid. The motion caused by the rotor unbalance is similar to the case when the cavity is completely filled with liquid.

**Author Contributions:** Conceptualization, A.K.; methodology, A.K.; software, G.-G.A.I; validation, A.Z. and A.K.; formal analysis, A.K. and A.Z.; investigation, A.K. and A.Z.; resources, A.K. and A.Z.; data curation, G.-G.A.I., A.Z. and A.K.; writing—original draft preparation, A.K.; writing—review and editing, A.Z.; visualization, G.-G.A.I.; supervision, A.Z.; project administration, A.K.; funding acquisition, A.K. All authors have read and agreed to the published version of the manuscript.

**Funding:** This research was funded by Al-Farabi Kazakh National University, grant number AP19677384.

**Institutional Review Board Statement:** This study did not require ethical approval.

**Informed Consent Statement:** Not applicable.

**Data Availability Statement:** Data are contained within the article.

**Acknowledgments:** This work has been supported financially by the research project (AP19677384—Development and research of the dynamics of a gas centrifuge on magnetic bearings with nonlinear characteristics and a control system) of the Ministry of Science and Higher Education of the Republic of Kazakhstan and was performed at the Research Institute of Mathematics and Mechanics in Al-Farabi Kazakh National University, which is gratefully acknowledged by the authors.

**Conflicts of Interest:** The authors confirm that they have no conflict of interest with respect to the work described in this manuscript.

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
