# Peer review of "Parametric Analysis of Nonlinear Oscillations of the “Rotor–Weakly Conductive Viscous Fluid–Foundation” System under the Action of a Magnetic Field"

_applsci, doi:10.3390/app132112089_

Round 1

Reviewer 1 Report

Please find my review below.

Extensive english editing is required. As i have mentioned in my comments, the sentence structuring is way off, leading to long sentences spanning 4 to 5 lines. 

Author Response

Response to Reviewer’s Comments

ID applsci-2645580: Parametric Analysis of Nonlinear Oscillations of the “Rotor-Weakly Conductive Viscous Fluid-Foundation” System under the Action of a Magnetic Field

Response to Reviewer А:

Comment 1

Extensive editing of English language needs to be done.

The sentence structuring is off which leads to confusing usage of commas, which makes the sentences long and is very difficult for readers to understand. In some cases, sentences span over 4 lines long, such as the following from line 78 to 83. “The solution to this problem is complicated by the fact that the motion of a rotating rotor and the motion of a weakly conductive viscous fluid in its cavity are interconnected under the action of the electromagnetic field, which causes a change in the frequency of forced oscillations and instability [49], and the system being solved includes the equations of movement of a solid body, the equations of a continuous medium and boundary conditions for the liquid [50]”

The example cited above is just one of the many different cases I have seen through out the paper. The authors need to thoroughly go through the paper and restructure the long sentences used throughout the paper.

Response 1

The authors took this comment into account and carried out appropriate work. For convenience, edits have been highlighted.

Comment 2

Line 378, there is a typo in the equation.

Response 2

The typo has been corrected.

Comment 3

Line 423 to 425, From the earlier and the last formulas [37], it is obvious that the amplitudes of nat- 423 ural oscillations of the rotor B and foundation D depend on the natural frequency of 424 the nonlinear system [38-40]. Did the authors meant to cite the papers or cross reference the equation?

Response 3

In this case, the works of other authors are cited. Thus, the authors wanted to once again emphasize the peculiarity of nonlinear systems.   

Comment 4

The authors are expected to define critical speed, ? in line 294

Response 4

In this case, the authors build amplitude-frequency characteristics, which is no less important from a practical point of view. A system of algebraic equations for determining and constructing amplitude-frequency characteristics was obtained by substituting a solution in the form (16) and (17) into differential equations of motion of the form (30) and further non-dimensionalizing it in the form (32), where ? is the dimensionless natural frequency system, while s is the dimensionless angular velocity of the rotor.

Comment 5

In fig 2, since all lines are on top of each peaks, its difficult for readers to understand where each one of the lines fall. Authors are expected to show a zoomed up portion of one or all of the peaks in the same picture. Similar to Fig 6

Response 5

Enlarged parts of these areas have been added to Figures 2 and 3.

Comment 6

In line 294, what is n0? Is there a physical meaning of n0? Does it represent RPM?

Response 6

This designation n0 does not represent the number of revolutions per minute and was introduced to simplify the final form of algebraic equations (32a)-(35).

Comment 7

Figure 6 and 7, labels are all in a foreign language.

Response 7

This typo appeared during document conversion. The error has been fixed. Thank you for your comment.

Comment 8

The authors are expected to show a figure of gamma on x axis ad various amplitudes of first, second and third critical speeds on y axis, with subplots for bm. This will help readers have a better understanding of the effect of gamma.

Response 8

Work in this direction has already been done in such works as

  1. Saito Y., Sawada T. Liquid sloshing in a rotating, laterally oscillating cylindrical container //Universal Journal of Mechanical Engineering. – 2017. – V. 5. – No. 3. – pp. 97-101;
  2. Ishida, Y., & Yamamoto, T. (2013). Linear And Nonlinear Rotordynamics: a modern treatment with applications. John Wiley & Sons.
  3. Moiseev N.N. The motion of a body having cavities partially filled with an ideal dropping liquid. - DAN USSR. V. 85. No. 4. 1952;
  4. V. Derendyaev and V.M. Sandalov. “On the stability of the stationary rotation of a cylinder partially filled with a viscous incompressible fluid.” Applied Mathematics and Mechanics. - 1982. - V. 46. - No. 4. - pp. 578-586;
  5. Galdi G. P., Mazzone G. and Zunino P. Inertial Motions of a Rigid Body with a Cavity Filled with a Viscous Fluid // Comptes Rendus Mecanique. — 2013. — Vol. 341. No 11–12. — Pp. 760-765;
  6. Kydyrbekuly A., Ibrayev G. G. A., Ospan. T., Nikonov A. (2021). Multi-parametric Dynamic Analysis of a Rolling Bearings System. Strojniski Vestnik/Journal of Mechanical Engineering. 67(9).

In this connection, the authors decided not to repeat the analysis on the effect of γ, i.e. filling the cavity with liquid on the amplitudes of the system as a whole.

Reviewer 2 Report

Manuscript ID: applsci-2645580

Title: Parametric Analysis of Nonlinear Oscillations of the “Rotor-Weakly Conductive Viscous Fluid-Foundation” System under the Action of a Magnetic Field

This manuscript presents an analytically generalized dynamic model of the “rotor-weakly conductive fluid-foundation” system, which takes into account the electromagnetic properties of a viscous fluid and its vibrations, the nonlinear stiffness properties of bearing supports and vibrations of the foundation, was developed and solved. The topic is interesting, and the manuscript is well organized. However, the reviewer has some comments below.

1.      While the author presented a comprehensive theoretical analysis of the system. Can the author discuss the feasibility of the experiment for this work?

2.      The reference works in the introduction part and reference list are too old.

3.      The author should refer to and compare the achieved results with some current studies.

4.      The quality of the Figures is poor and needs to be enhanced (Figs (2-11)).

5.      Translate Figs (6-7) to English.

Moderate editing of English language is required.

Author Response

Response to Reviewer’s Comments

ID applsci-2645580: Parametric Analysis of Nonlinear Oscillations of the “Rotor-Weakly Conductive Viscous Fluid-Foundation” System under the Action of a Magnetic Field

Response to Reviewer B:

This manuscript presents an analytically generalized dynamic model of the “rotor-weakly conductive fluid-foundation” system, which takes into account the electromagnetic properties of a viscous fluid and its vibrations, the nonlinear stiffness properties of bearing supports and vibrations of the foundation, was developed and solved. The topic is interesting, and the manuscript is well organized. However, the reviewer has some comments below.

Comment 1

While the author presented a comprehensive theoretical analysis of the system. Can the author discuss the feasibility of the experiment for this work?

Response 1

Experimental work in a similar setting was carried out in the work

Urbiola-Soto, L.; Lopez-Parra, M. Liquid self-balancing device effects on flexible rotor stability. Shock and Vibration, 2013, 20(1), 109-121.

Conducting experimental work for this work would undoubtedly improve the amplitude prediction process for rotor installations of this type. For example, separators and centrifuges often work with low-flowing liquid. The most complete consideration of the physical characteristics of the liquid would allow a more optimal approach to the issue of designing such rotary machines, which would undoubtedly be more profitable from an economic point of view. Unfortunately, at the moment, it is not possible for the authors to carry out experimental work in exactly the same setting.

Comment 2

The reference works in the introduction part and reference list are too old.

Response 2

This work in this direction is quite specific, and the authors made every effort to select the most modern similar works as possible.

Comment 3

The author should refer to and compare the achieved results with some current studies.

Response 3

The case in the absence of a magnetic field was considered in sufficient detail in such works as

  1. Saito Y., Sawada T. Liquid sloshing in a rotating, laterally oscillating cylindrical container //Universal Journal of Mechanical Engineering. – 2017. – V. 5. – No. 3. – pp. 97-101;
  2. Ishida, Y., & Yamamoto, T. (2013). Linear And Nonlinear Rotordynamics: a modern treatment with applications. John Wiley & Sons.
  3. Moiseev N.N. The motion of a body having cavities partially filled with an ideal dropping liquid. - DAN USSR. V. 85. No. 4. 1952;
  4. V. Derendyaev and V.M. Sandalov. “On the stability of the stationary rotation of a cylinder partially filled with a viscous incompressible fluid.” Applied Mathematics and Mechanics. - 1982. - V. 46. - No. 4. - pp. 578-586;
  5. Galdi G. P., Mazzone G. and Zunino P. Inertial Motions of a Rigid Body with a Cavity Filled with a Viscous Fluid // Comptes Rendus Mecanique. — 2013. — Vol. 341. No 11–12. — Pp. 760-765;
  6. Kydyrbekuly A., Ibrayev G. G. A., Ospan. T., Nikonov A. (2021). Multi-parametric Dynamic Analysis of a Rolling Bearings System. Strojniski Vestnik/Journal of Mechanical Engineering. 67(9).

There are no works by other authors in exactly the same setting as in this work. Thus, this article covers the results of previous work.

Comment 4

The quality of the Figures is poor and needs to be enhanced (Figs (2-11)).

Response 4

The quality of figures 2-11 has been improved.  

Comment 5

Translate Figs (6-7) to English.

Response 5

This bug has been corrected.
